# Symptomatic dry eye disease (DED) in cohort of contact lens wearers in Jordan

**Wissam Ghach**[ID][1]*, **May M. Bakkar**[2]*, **Mona Aridi**[ID][3], **Mohammad A. Alebrahim**[ID][2]

**1** Department of Public Health, Canadian University Dubai, Dubai, United Arab Emirates, **2** Faculty of Applied Medical Sciences, Jordan University of Science and Technology, Irbid, Jordan, **3** Univ Angers, LARIS, SFR MATHSTIC, Angers, France

* bakkar.may@gmail.com (MMB); Wissam.ghach@cud.ac.ae (WG)

## Abstract

Understanding the symptomatic dry eye disease (DED) and its associated risk factors among contact lens wearers is crucial for clinicians to tailor effective interventions, enhance patient care, and prevent contact lens dropout. This study investigated symptomatic DED and its associated risk factors among a sample of contact lens wearers in Jordan. This cross-sectional study assessed symptomatic DED in a cohort of contact lens wearers in Jordan using an online survey distributed across various social media platforms. A total of 301 participants completed the survey, which included demographic and contact lens profile questions and the Arabic version of the Ocular Surface Disease Index (ARB-OSDI) questionnaire. Statistical analyses explored the associations between OSDI scores, demographics, symptoms, visual-related functions affected by dryness, and triggers of dryness. Among the study population, 77.1% were females, 48.2% were aged 18–24 years old, and 24.87% were soft contact lens wearers. The mean OSDI score was 22.9±17, with 70% showing mild-to-severe dry symptoms and 25% showing severe symptomatic DED. The ANOVA revealed a significant association between symptomatic DED, wearing face masks, longer contact lens age, and poor cleaning habits. The use of lubricant eye drops significantly reduced symptomatic DED with a mean OSDI score of 8.79. The most prevalent dryness symptoms were pain and blurred vision, affecting reading and TV watching in 50% of the population. Wind and air conditioning were the most common environmental triggers, reported by 67.8% and 66.4% of participants, respectively. A high proportion of symptomatic DED was reported in this study population. Wearing face masks, a longer contact lens age, and poor contact lens hygiene were correlated with exaggerated DED symptoms. Conversely, the use of lubricated eye drops reduces the symptoms of DED.

**Data availability statement:** All relevant data are within the paper and its Supporting Information files.

**Funding:** This study protocol has received a grant from Deanship of Research at Jordan University of Science and Technology - Research Grant No: 20230271. Acknowledgment is also given to the Canadian University Dubai (CUD) and Jordan University of Science and Technology (JUST) for approving and supervising the study design and protocol.

**Competing interests:** The authors have declared that no competing interests exist.

**Abbreviations:** DED, dry eye disease; OSDI, ocular surface disease index; AC; air-conditioning.

## 1. Introduction

Dry eye disease (DED) is a common multifactorial disorder of the ocular surface that can significantly affect vision and comfort. The Tear Film and Ocular Surface Dry Eye Workshop II (TFOS DEWS II) defines DED as "multifactorial disease of the ocular surface characterized by a loss of homeostasis of the tear film and accompanied by ocular symptoms, in which tear film instability and hyperosmolarity, ocular surface inflammation and damage, and neurosensory abnormalities play etiological roles'' [1,2].

Contact lens wear has been listed as a modifiable risk factor for DED, with contact lens–related dryness often cited as a leading cause of discomfort, reduced wearing time, and discontinuation of lens use [3]. Estimates suggest that between 15% and 55% of contact lens wearers experience symptoms consistent with dry eye, a rate higher than that observed in non–contact lens wearers [4–9]. Suggested mechanisms of contact lens-related dry eye include increased tear film instability caused by contact lens friction with the ocular surface, accelerated pre-corneal tear film evaporation and subsequent tear film thinning, reduced contact lens wettability, ocular surface inflammation, and meibomian gland dysfunction [10].

Symptoms of contact lens-related DED include dryness, reduced vision quality, foreign body sensation, eye strain, blurred vision, ocular discomfort, and contact lens intolerance [10] can be assessed using available validated dry eye symptomology questionnaires. These include the Ocular Surface Disease Index (OSDI), MacMonnies' Questionnaire, Standard Patient Evaluation of Eye Dryness (SPEED), or specific questionnaires developed for use with contact lens wearers such as the 8-items Contact Lens Dry Eye Questionnaire (CLDEQ-8) [11].

In Jordan, population-based studies have shown that the prevalence of DED symptoms is relatively high. Prior to the COVID-19 pandemic, the reported percentage of individuals experiencing mild-to-severe DED symptoms was 59% [12], and this figure increased to 73.4% during the pandemic [13]. However, these investigations evaluated the general population without specifically examining contact lens wearers.

The current study aimed to investigate the proportion of contact lens wearers in Jordan reported symptomatic DED during the COVID-19 pandemic, using the validated Arabic version of the OSDI (ARB-OSDI) questionnaire. In addition, potential risk factors, including age, sex, frequency of mask use, contact lens type, duration of contact lens wear, and contact lens care practices) were examined to evaluate their statistical association with DED symptoms. The findings of this study aim to raise awareness of symptomatic DED in contact lens wearers and support the development of strategies to reduce discomfort and minimize contact lens discontinuation rates in the region.

## 2. Materials and methods

### 2.1. Study design and population

A cross-sectional study design is utilized to assess DED symptoms among a cohort of contact lens wearers in Jordan. The study population was recruited between

15/05/2022 and 30/11/2022 using convenience sampling. A total of 301 participants met the inclusion criteria and successfully completed the survey where only contact lens wearers aged ≥ 18 years participated in the study. Recruitment process included a random distribution of the Google form questionnaire survey distributed on several social media platforms (Facebook, Instagram, Twitter, and LinkedIn) to reach a large sample of contact lens wearers among social media users in Jordan. Participants with a history of eye surgeries, active ocular diseases, or use of ocular or systemic medications (except for lubricant eye drops) known to interfere with tear film production, or ocular surface integrity were excluded from the study. Examples of such conditions and medications were provided in the questionnaire.

## 2.2. Study tool

The questionnaire used in the survey included two sections. The first covered participants' profile characteristics, including gender, age, contact lens type, duration of contact lens wear (in hours), total duration of contact lens use since first fitting (in months), compliance with contact lens cleaning and lubricant eye drop use, and mask use during COVID-19 (mask use alone or in combination with contact lens wear, and duration of mask use per day). Mask use data were self-reported, with participants asked to report on their typical mask-wearing habits during the three months preceding the survey. The second section consisted of the validated Arabic version of the OSDI questionnaire to assess the severity of dryness symptoms induced by environmental factors over the preceding week [14]. The OSDI questionnaire was originally created by the Outcomes Research Group at Allergan Inc. (Irvin, California, USA) [15] to quantify the prevalence of DED. The 12-item OSDI questionnaire consisted of three sections: five questions about ocular symptoms, four questions about vision-related functions, and three questions about environmental triggers. Each item was scored on a scale of 0–4, where 0 indicated none of the time; 1, some of the time; 2, half of the time; 3, most of the time, and 4 indicated all of the time [15]. Then, each individual OSDI score was calculated using the following formula.

$$OSDI\ Score = \frac{Sum\ of\ scores\ for\ all\ questions\ answered\ \times 100}{Total\ number\ of\ questions\ answered\ \times 4}$$

Individual OSDI scores, each ranging from 0 to 100, were used to calculate the mean OSDI score, with higher values reflecting greater disability.

Dry eye symptoms were assessed using OSDI scores, with a cut-off of ≥13 distinguishing normal from symptomatic DED, which was further classified as mild, moderate, or severe.

The measurement scale of the OSDI status was divided into three intervals: the interval of [0–12] represents normal cases of non-dry eye, the interval of [13–32] represents mild-to-moderate dry eye, and the interval of [33–100] represents severe dry eye [13,16].

## 2.3. Data analysis

Statistical Package for the Social Sciences version 21 (SPSS, International Business Machine Corp. IBM, Chicago, IL, USA) was used for data analysis. Descriptive analyses based on frequency and percentage distributions were performed for all the variables. The percentage of the population with an OSDI score greater than 13 was used to determine the proportion of symptomatic DED (mild to moderate and severe OSDI status). One-way analysis ANOVA and cross-tabulation tests were used to assess significant differences across the studied variables. The Pearson's correlation test was used to check for correlations across the studied variables. The level of statistical significance was set at $p < 0.05$. Before performing One-way ANOVA, the assumptions of normality and homogeneity of variances were tested. The Shapiro-Wilk test was used to assess data normality. Subsequently, the Levene's test was performed to evaluate the homogeneity of variances across groups. One-way ANOVA was performed to compare the OSDI scores across the different groups only if both assumptions were met.

Potential confounding factors were identified from prior literature on dry eye disease and contact lens–related discomfort. The following variables were considered potential confounders and were included as independent predictors in the multiple linear regression analysis to adjust for their effects: age, gender, history of contact lens–mask interaction (CL_Mask), years of contact lens use (CL_use_year), mask-wearing status (Do_you_use_mask), mask-wearing frequency (Mask_use_frequency), presence of pre-existing contact lens–related symptoms (CL_symptoms), type of contact lenses worn (CL_types), daily contact lens use (CL_use_daily), contact lens cleaning rate (CL_cleaning_rate), and use of eye drops (Eyedrop_use). This multivariable approach was employed to estimate the independent association between each predictor and OSDI scores while minimizing the influence of confounding.

To account for potential confounding factors and to identify independent predictors of symptomatic dry eye severity, a multiple linear regression analysis was conducted with the continuous OSDI score as the dependent variable. Variables included in the regression model were selected based on theoretical relevance and prior univariate analysis. Statistical significance was set at $p < 0.05$ for all analyses.

## 2.4. Ethical considerations

The Institutional Review Boards (IRB) of Jordan University of Science and Technology and Modern University for Business and Science reviewed and approved the study protocol (MU-20210323-22E). This study was conducted in accordance with the principles of the Declaration of Helsinki. All participants provided written informed consent electronically prior to their participation in the study. Confidentiality was maintained during data collection and processing.

# 3. Results

## 3.1. Profile characteristics of contact lens wearers

A total of 301 respondents successfully participated in this study. The majority of the respondents (77.1%) were female, with a gender ratio (F: M = 3.36:1), and aged between 18 and 44 years old (96%). Table 1 shows the descriptive statistics of the characteristics of the contact lens wearers who participated in this study. Most of the participants (69.7%) had mild-to-severe symptomatic DED (OSDI score was ≥ 13), with a mean OSDI score of $22.9 \pm 17$.

To evaluate the normality assumption for ANOVA, a **Shapiro-Wilk test** was conducted on the dependent variable, the OSDI score (Table S1 in S2 File). The test statistics were 0.946 with a p-value of 0.081, indicating that the data did not deviate significantly from normality ($p > 0.05$). Therefore, the assumption of normality was satisfied, and an ANOVA was conducted.

Additionally, **Levene's test** for the homogeneity of variances showed no significant violations of the homogeneity assumption ($p > 0.05$) for all grouping variables (Table S2 in S2 File). Therefore, one-way ANOVA was performed to compare mean OSDI scores across the groups. To investigate the statistical association and correlation of the DED risk factors among the contact lens wearers, One-way ANOVA and Pearson correlation tests were carried out on the OSDI score of the severity categories (normal, mild to moderate, and severe) with respect to profile characteristics of the study population as shown in Table 2.

To assess the difference in the mean OSDI score across several categorical variables, the dependent variable OSDI was tested against the following independent variables: sex, age group, type of contact lens, contact lens duration per day, contact lens age, cleaning rate of the contact lens, use of mask during contact lens use, and frequency of mask use per day. Similarly, a chi-square test was conducted to evaluate the relationships between the tested variables and OSDI scores. Statistically significant difference was considered when P-value is less than 0.05. The significant p-values are shown in Table 2. The statistical analysis of OSDI severity categories with respect to age showed that the age group of "25-44 years old" had the lowest severity of symptomatic DED and the lowest mean score, with no significant difference ($P > 0.05$) among all age groups. However, age was significantly correlated ($P < 0.05$) with OSDI severity categories. On the other hand, the statistical analysis of OSDI severity categories with respect to sex showed no significant differences

**Table 1. Frequency and percent frequency of Profile characteristics (gender, age, contact lens type, contact lens age, frequency of lubricant eye drops use, cleaning rate of contact lenses, use of mask during contact lens wear, frequency of mask use per day, and OSDI status) of the contact lens wearers (n = 301).**

| Variables | | Frequency (%) |
|---|---|---|
| **Gender** | Male | 69 (22.9%) |
| | Females | 232 (77.1%) |
| **Age** | 18-24 years old | 145 (48.2%) |
| | 25-44 years old | 144 (47.8%) |
| | ≥ 45 years old | 12 (4%) |
| **Contact lens type** | Soft | 262 (87%) |
| | Rigid gas permeable (RGB) | 39 (13%) |
| **Contact lens wear duration per day (in hours)** | Less than 12 hours | 265 (88%) |
| | More than 12 hours | 36 (12%) |
| **Contact lens age (in months)** | Less than 6 months | 68 (22.6%) |
| | 7–12 months | 123 (40.8%) |
| | More than 12 months | 110 (36.6%) |
| **Frequency of lubricant eye drops use** | Never | 84 (28%) |
| | Sometimes | 107 (35.5%) |
| | Always | 110 (36.5%) |
| **Cleaning rate of contact lenses** | Never | 87 (29%) |
| | Sometimes | 98 (32.5%) |
| | Daily | 116 (38.5%) |
| **Use of a mask during contact lens wear** | No | 135 (44.9%) |
| | Yes | 166 (55.1%) |
| **Frequency of mask use per day** | 0-3 hours | 162 (53.8%) |
| | 4-6 hours | 75 (24.9%) |
| | More than 6 hours | 64 (21.3%) |
| **DED status based on OSDI scores** | Normal (OSDI score: 0–12) | 90 (30.3%) |
| | Mild-to-Moderate (OSDI score: 13–32) | 134 (44.3%) |
| | Severe | 77 (25.7%) |

or correlations between male and female participants (P > 0.05). Interestingly, the respondents who wore contact lenses for more than 12 months had never cleaned their contact lenses and had never used lubricant eye drops had the highest mean OSDI scores with significant differences (p < 0.05).

The Pearson correlation test supported the hypothesis that the longer the contact lens age (in months), the higher the severity of the symptomatic DED (p < 0.001). Similarly, Pearson analysis confirmed that participants who reported poor compliance with contact lens hygiene and who did not use lubricant eye drops had a higher severity of symptomatic DED (p < 0.001). In contrast, the type of contact lenses and contact lens duration per day showed no significant difference and statistical correlation with symptomatic DED represented by the OSDI mean score and severity categories, respectively (P > 0.05).

Focusing on the mask-related variables, the respondents who wore a mask along with their contact lenses, specifically for more than six hours per day, had the highest mean scores of OSDI with significant differences (p < 0.05). According to the Pearson correlation test, wearing a mask along with contact lenses was found to be statistically correlated (p < 0.001) with the severity of symptomatic DED represented by the OSDI mean score. Additionally, the correlation test confirmed the hypothesis that the longer the use of masks and contact lenses, the higher the severity of symptomatic DED (p < 0.001).

**Table 2. Statistical association (One-Way ANOVA) and correlation (Pearson) among OSDI status (Normal, Mild-toModerate, Severe), OSDI mean scores, and profile characteristics (gender, age, contact lens type, contact lens age, frequency of lubricant eye drops use, cleaning rate of contact lenses, use of mask during contact lens wear, frequency of mask use per day) of the contact lens wearers (301 participants).**

| Variable | | Normal | Mild-to-Moderate | Severe | OSDI Mean Score (SD) |
|---|---|---|---|---|---|
| | | **Frequency (percentages)** | | | |
| **Gender** | **Male** | 23(33.3%) | 27(39.1%) | 19(27.5%) | 22.61(17.43) |
| | **Female** | 67(28.9%) | 106(45.7%) | 59(25.4%) | 23.12(17.03) |
| One-Way ANOVA | | F = 0.047; p=0.829 | | | |
| Pearson Correlation | | $\chi^2$ =0.952; p=0.618 | | | |
| **Age** | **18-24 years old** | 46(31.7%) | 51(35.2%) | 48(33.1%) | 24.44(18.51) |
| | **25-44 years old** | 43(29.9%) | 74(51.4%) | 17(18.8%) | 21.47(15.80) |
| | **≥ 45 years old** | 1(8.3%) | 8(66.7%) | 3(25.0%) | 24.13(13.34) |
| One-Way ANOVA | | F = 1.118; p=0.328 | | | |
| Pearson Correlation | | $\chi^2$=13.506; p=0.009 | | | |
| **Contact lens type** | **Soft** | 75 (28.7%) | 119 (45.6%) | 67 (25.7%) | 23.44 (16.94%) |
| | **Rigid gas permeable (RGB)** | 15 (38.5%) | 13 (33.3%) | 11 (28.2%) | 19.98 (18.21%) |
| One-Way ANOVA | | F = 0.758; p=0.470 | | | |
| Pearson Correlation | | $\chi^2$ =7.797; p = 0.099 | | | |
| **Contact lens wear duration per day (in hours)** | **Less than 6 hours** | 37 (30.6%) | 60 (49.6%) | 24 (19.8%) | 22.50 (16.44%) |
| | **7–12 hours** | 44(30.3%) | 61(42.1%) | 40(27.6%) | 23.18 (19.99%) |
| | **More than 12 hours** | 9 (25.7%) | 12 (34.3%) | 14 (40%) | 27.20 (19.99%) |
| One-Way ANOVA | | F = 0.805; p=0.491 | | | |
| Pearson Correlation | | $\chi^2$ =12.179; p=0.058 | | | |
| **Contact lens age (in months)** | **Less than 6 months** | 44 (64.7%) | 20 (29.4%) | 4 (5.9%) | 10.91 (11.47%) |
| | **7–12 months** | 28 (22.8%) | 81 (65.9%) | 14 (11.4%) | 20.53 (11.89%) |
| | **More than 12 months** | 18 (27.3%) | 22 (33.3%) | 26 (39.4%) | 26.23 (18.28%) |
| One-Way ANOVA | | **F = 39.750; p < 0.001** | | | |
| Pearson Correlation | | **$\chi^2$=142.345; p < 0.001** | | | |
| **Frequency of lubricant eye drops use** | **Never** | 1 (1.2%) | 17 (20.5%) | 65 (78.3%) | 42.37 (13.97%) |
| | **Sometimes** | 9 (8.4%) | 87 (81.3%) | 11 (10.3%) | 22.55 (9.51%) |
| | **Always** | 80 (72.7%) | 28 (25.5%) | 2 (1.8%) | 8.79 (8.92%) |
| One-Way ANOVA | | **F = 154.410; p < 0.001** | | | |
| Pearson Correlation | | **$\chi^2$=279.553; p < 0.001** | | | |
| **Cleaning rate of contact lenses** | **Never** | 0 (0%) | 21 (24.4%) | 65 (75.6%) | 40.99 (13.57%) |
| | **Sometimes** | 3 (3.1%) | 85 (86.7%) | 10 (10.2%) | 24.11 (10.61%) |
| | **Daily** | 87 (75%) | 26 (22.4%) | 3 (2.6%) | 8.69 (9.03%) |
| One-Way ANOVA | | **F = 142.520; p < 0.001** | | | |
| Pearson Correlation | | **$\chi^2$=301.620; p < 0.001** | | | |
| **Use of mask during contact lens wear** | **No** | 89 (54.6%) | 73 (44.8%) | 1 (0.6%) | 10.94 (8.34) |
| | **Yes** | 1 (0.7%) | 60 (43.5%) | 77 (55.8%) | 37.26 (13.41) |
| One-Way ANOVA | | **F = 431.020; p < 0.001** | | | |
| Pearson Correlation | | **$\chi^2$=163.853; p < 0.001** | | | |
| **Frequency of mask use per day** | **0–3 hours** | 88 (54.3%) | 73 (45.1%) | 1 (0.6%) | 11.12 (8.58) |
| | **3–6 hours** | 2 (2.7%) | 53 (73.3%) | 18 (24%) | 28.03 (8.13) |
| | **> 6 hours** | 0 | 5 (7.8%) | 59 (92.2%) | 47.20 (11.75) |
| One-Way ANOVA | | **F = 405.410; p < 0.001** | | | |
| Pearson Correlation | | **$\chi^2$=761.664; p < 0.001** | | | |

*Values in bold indicate P < 0.05.

### 3.2. Analysis of dryness symptoms

The most frequent dryness symptoms were pain and blurred vision, where 62.1% and 60.1% of the study population reported these symptoms on the rate some of the time to all the time, respectively, Fig 1).

### 3.3. Analysis of impact of dryness symptoms on vision-related functions

Fig 2 shows the effect of dryness on vision-related functions. Reading and TV watching were the most affected vision-related functions (from "some of the time" to "all the time") among 53.5% and 51.5% of the study population, respectively. However, night driving and computer use were the least affected activities among the contact lens wearers.

### 3.4. Analysis of environmental triggers

Windy conditions and areas that Air conditioners emerged as prominent environmental triggers for dryness symptoms, affecting 67.8% and 66.4% of the study population, respectively. In contrast, low humidity (very dry areas) had the least effect on the dryness symptoms reported by contact lens wearers (Fig 3).

### 3.5. Multiple linear regression analysis

A multiple linear regression model was fitted to examine the independent associations between demographic and contact lens–related factors and Ocular Surface Disease Index (OSDI) score as a continuous outcome. As presented in Table S3 in S2 File, the overall model was statistically significant (**F** = 126.245, $p < 0.001$) and demonstrated a strong explanatory power, accounting for **82.8%** of the variance in OSDI scores ($R^2 = 0.828$, adjusted $R^2 = 0.821$). Inspection of the residual plots confirmed that the assumptions of linearity, homoscedasticity, and normality of residuals were met. However, multi-collinearity diagnostics indicated that all predictors had variance inflation factor (VIF) values > 5 and tolerance values < 0.2, suggesting a high degree of collinearity among the independent variables.

After adjusting all variables in the model, several predictors were significantly associated with OSDI score. Longer contact lens use in years (B = 2.466, $p < 0.001$) and higher frequency of mask use (B = 9.745, $p < 0.001$) were strong positive predictors, indicating that each additional year of lens use and greater mask-wearing frequency were associated with higher OSDI scores, reflecting more severe dry eye symptoms. Similarly, reporting the presence of contact lens–related symptoms (B = 2.234, $p < 0.001$) was associated with a significant increase in OSDI score.

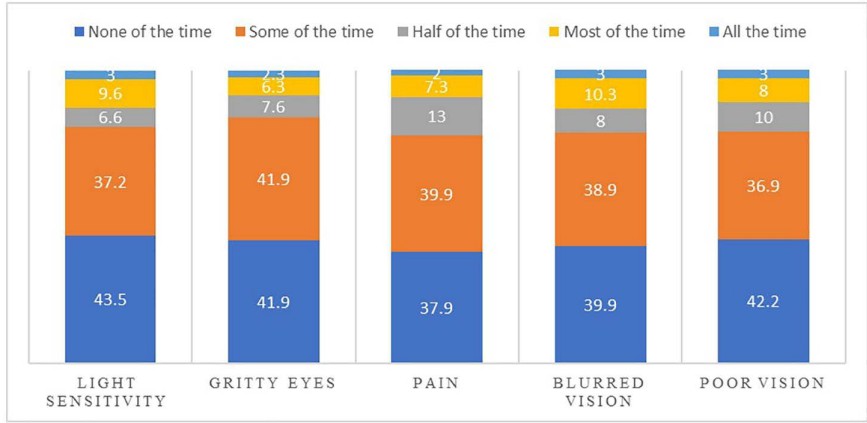

**Fig 1. Bar graph represents the percentage frequencies of ocular symptoms reported by the study population.**

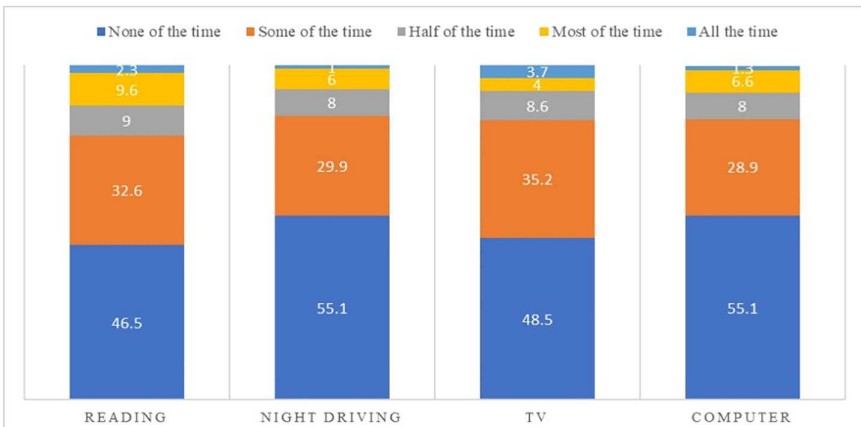

**Fig 2. Bar graph representing the percentage frequencies of impact of dryness symptoms on vision-related functions by the sample population.**

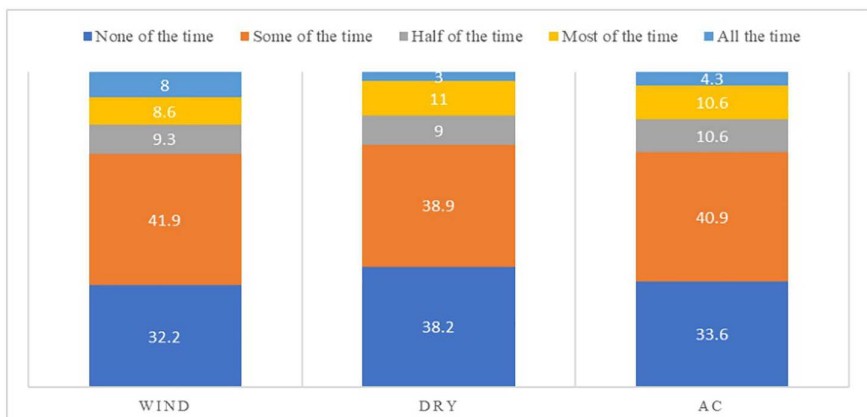

**Fig 3. Bar graph representing the percentage frequencies of different environmental triggers to DED symptoms by the sample population.**

Conversely, higher rates of compliance with contact lens cleaning (B=−4.445, $p<0.001$) and the use of lubricant eye drops (B=−4.430, $p<0.001$) were independently associated with lower OSDI scores, suggesting a protective effect against dry eye symptoms. Other factors, including age, gender, mask use as a binary variable, contact lens type, daily hours of lens wear, and simultaneous mask and contact lens use, were not statistically significant predictors in the adjusted model ($p>0.05$ for all).

Among the standardized coefficients (β), the frequency of mask use (β=0.459) emerged as the strongest predictor, followed by contact lens cleaning rate (β=−0.214), lubricant eye drop use (β=−0.209), contact lens–related symptoms (β=0.124), and years of contact lens use (β=0.112).

The histogram of the regression standardized residuals was roughly bell-shaped, Fig S1 in S2 File, indicating that the residuals were approximately normally distributed, which supports the assumption of normality. Additionally, the scatter plot of the standardized predicted values against the standardized residuals displayed a random, Fig S2 in S2 File, evenly dispersed pattern without any clear curvature or systematic structure. This suggests homoscedasticity of residuals and

confirms that the variance of errors is constant across predicted values, fulfilling one of the key assumptions of linear regression.

## 4. Discussion

Symptomatic DED has been extensively reported in contact lens-wearers. The DED symptoms associated with contact lens wear can vary depending on various factors, such as contact lens-related factors (e.g., contact lens type, contact lens material and design, wear modality, replacement schedule, and duration of contact lens wear), mask-related factors (e.g., mask wear and frequency of use with/without contact lenses), environmental factors (e.g., low humidity and exposure to higher temperatures), and patient-related factors (e.g., gender, age, poor contact lens wear compliance, and concurrent ocular surface conditions [10,17–19].

The contact-lens market in Jordan is evolving. Haddad et al. (2019) reported that the lens prescribing trend in Jordan is in line with global data in terms of lens material, fit type, design, cosmetic lens use, and preference for frequent replacement lenses. However, the procurement of presbyopia and orthokeratology contact lenses remains suboptimal [20]. In the current study, the majority of the participants were females and wore soft contact lenses on a frequent replacement basis. This finding is in agreement with that of a previous report on contact lens prescription trends in Jordan [20].

### 4.1. Symptomatic DED in the study population

This study aimed to explore DED symptoms among a cohort of contact lens wearers in Jordan, utilizing the ARB-OSDI and employing a cut-off value of OSDI ≥13. Additionally, this investigation aimed to study the factors associated with DED symptomatology within the study sample. This study was conducted during the COVID-19 pandemic, during which the mandatory use of face masks was enforced in public and workplace settings.

The study also revealed that among contact lens wearers, the majority experienced mild-to-severe symptomatic DED, with an overall mean OSDI score indicating notable symptoms.. This finding corresponds with a prior study conducted in Jordan in 2016 by Bakkar *et al*, which reported a symptomatic DED prevalence of 70% among contact lens wearers. However, the primary focus of this study was to determine the prevalence of DED symptoms and its associated risk factors, including contact lens wear, in a non-clinical population in Jordan [12]. Additionally, this study employed a smaller sample size of 110 contact lens wearers.

The current study found that proportion of symptomatic DED among contact lens wearers is found to be higher than the prevalence rate of DED among contact lens wearers in the literature, which they demonstrated a considerable variability ranging from 15% to 55% [4–9]. This variability in the literature can be attributed to the characteristics of the study population, differences in the types of contact lens materials used, and differences in diagnostic criteria based on the presence of symptoms, clinical signs of dryness or both [10].

### 4.2. Risk factors of symptomatic dry eye disease

This study also explored the correlations between different OSDI severity categories and predetermined risk factors. The study showed that age was significantly correlated with OSDI severity categories. This is in accordance with other studies suggesting that aging is a predictor of symptomatic DED [21].

The results also showed no significant difference between the different OSDI severity categories and participants' gender. However, this finding contradicts other reports that found that the proportion of symptomatic DED in female contact lens wearers was higher than that in male contact lens wearers [10]. The limited representation of male contact lens wearers in the current study may justify this.

Interestingly, the study showed that contact lens wearers who wore their lenses for a period exceeding 12 months and who were non-compliant with the contact lens cleaning regimen had the highest mean OSDI score with significantly higher severity of symptomatic DED. This may be explained by the accumulation of lens deposits, encompassing lipids and proteins, on the surface of the contact lenses during extended lens usage, particularly in conjunction with poor lens hygiene. Lens deposits may alter tear film quality and reduce contact lens wettability, factors that are likely to contribute to the onset of DED symptoms during prolonged contact lens wear [22–25].

In contrast, other contact lens-related factors, such as types of contact lenses, that is, soft versus rigid gas permeable lenses, and contact lens wear duration per day, failed to demonstrate a statistically significant difference in the severity of symptoms associated with dryness. The difference in sample size between the groups wearing soft and rigid contact lenses and the groups in terms of daily wear duration may have contributed to the observed outcomes. An increase in the sample size could enhance the ability of this study to detect subtle effects.

The findings show that contact lens wearers who reported always using lubricant eye drops as part of their ocular lubrication regimen had a lower proportion of symptomatic DED, reflected by a reduced mean OSDI score within the normative range. While this observation suggests a potential association between the use of non-preserved lubricant eye drops and reduced dryness severity, causality cannot be inferred. Previous studies have similarly reported that the application of preservative-free lubricating eye drops before and after contact lens wear is associated with improvements in symptoms of contact lens–related dry eye [12,18,26–30].

The current study is distinctive in that incorporating a face mask for a prolonged duration during contact lens wear exacerbated self-reported dry eye symptoms (characterized by a significant increase in mean OSDI scores) when compared to short periods of mask wear. This finding is supported by many reports during the COVID-19 pandemic era that confirmed the existence of mask-associated dry eye (MADE) in noncontact lens wearers [19,31–38].

### 4.3. Dryness symptoms

In the present study, participants consistently reported symptoms of ocular dryness, including light sensitivity, gritty eyes, pain, blurred vision, and poor vision. In the study population, the most prevalent symptoms were pain and blurred vision, which were reported at frequencies ranging from some to all times. This finding markedly differs from the DED symptoms reported in a previous study conducted in Jordan, where it was found that "sensitivity to light" was found to be a more prevalent symptom [12]. This disparity suggests that further investigations are needed to explore the distinct prevalent dryness-related symptoms reported by contact lens wearers by studying the potential impact of contact lens interactions with the ocular surface on the manifestation of these particular symptoms.

### 4.4. Impact of dryness symptoms on vision-related functions

Among the vision-related functions affected by dryness symptoms, reading and watching TV were the most affected, as reported by the study sample. This outcome aligns with expectations, considering that the study was conducted during the COVID-19 era, characterized by lockdowns and prolonged periods of home quarantine that coincided with increased screen time due to online classes and remote work.

### 4.5. Environmental triggers for dryness symptoms

Among the environmental triggers, wind and air conditioning (AC) were reported by the majority of participants as the most common triggers for DED symptoms. These observations were based on self-reported data and were not statistically tested. The effects of both wind and AC—through dry and recirculated air—can reduce humidity in the surrounding environment, potentially increasing tear film evaporation and ocular surface dryness (40). Additionally, restricted outdoor activities

during the pandemic may have reduced exposure to natural elements such as wind and sunlight, potentially heightening sensitivity and intolerance to environmental conditions, which could contribute to increased dryness and discomfort (41).

### 4.6. Limitations of the study

This study has several limitations. First, reliance on an online survey introduced the potential for self-reporting bias and weakened the finding generalizability. Such a method disproportionately attracted younger, female, and more internet-active individuals, resulting in a skewed age gender distribution within the sample. Second, further limitation may arise from potential variability in participants' interpretation of the exclusion criteria, particularly concerning active ocular disease and the use of ocular or systemic medications that could affect ocular surface integrity. Although the study employed the OSDI questionnaire as a valid and reliable tool for identifying dryness symptoms, reliance on a single instrument may restrict the comprehensiveness of symptom assessment. Expanding the assessment of symptomatic DED using a designated contact lens dry eye questionnaire, such as the CLDEQ-8, along with additional clinical tests for DED could have enhanced diagnostic accuracy. This approach would not only confirm the diagnosis of symptomatic DED more precisely among contact lens wearers but also eliminate potential confounding factors associated with other DED-related conditions. Moreover, this study did not explore additional potential risk factors that could contribute to the development of DED, including the type of contact lens material, occupational influences, usage of video display terminals, screen time, and smoking habits, which may influence the development or severity of DED and should be considered in future research. Additionally, while the confounding variables were statistically adjusted for in the multivariable model, the possibility of residual confounding from unmeasured factors, such as environmental exposures, hormonal status, or undiagnosed ocular surface disease, cannot be excluded.

### 5. Conclusion

In conclusion, the results of this study suggest a high proportion (70%) of symptomatic DED among contact lens wearers in Jordan assessed by the validated Arabic version of the Ocular Surface Disease Index (OSDI). Symptoms of DED were associated with age, extended contact lens usage, suboptimal lens hygiene, and concurrent use of face masks during contact lens wear. Additionally, contact lens wearers, who consistently incorporate lubricant eye drops into their ocular lubrication routine, experience a significantly reduced proportion of symptomatic DED. These findings provide important baseline data for future research investigating behavioral and hygiene-related risk factors for symptomatic DED among the contact lens wearers in Jordan.

### Supporting information

**S1 File. An Excel sheet representing the coded data of the study population.**
(XLSX)

**S2 File. Supplementary Tables S1–S3, Figs S1 and S2.**
(DOCX)

### Acknowledgments

Acknowledgment is also given to the Canadian University Dubai (CUD) and Jordan University of Science and Technology (JUST) for approving and supervising the study design and protocol. Sincere gratitude is extended to all participants who contributed to this study.

### Author contributions

**Conceptualization:** Wissam Ghach, May M. Bakkar.

**Data curation:** Wissam Ghach, Mona Aridi, Mohammad A. Alebrahim.

**Formal analysis:** Mona Aridi.

**Methodology:** Wissam Ghach, May M. Bakkar, Mona Aridi, Mohammad A. Alebrahim.

**Project administration:** Wissam Ghach, May M. Bakkar.

**Supervision:** Wissam Ghach, May M. Bakkar.

**Writing – original draft:** Wissam Ghach, May M. Bakkar, Mona Aridi, Mohammad A. Alebrahim.

**Writing – review & editing:** Wissam Ghach, May M. Bakkar, Mohammad A. Alebrahim.

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
