## [Decision Letter · Decision Letter 0]

2 Aug 2025

Dear Dr. Ghach,

Thank you for submitting your manuscript to PLOS ONE. After careful consideration, we feel that it has merit but does not fully meet PLOS ONE’s publication criteria as it currently stands. Therefore, we invite you to submit a revised version of the manuscript that addresses the points raised during the review process.

We look forward to receiving your revised manuscript.

Kind regards,

Clara Martínez Pérez

Academic Editor

PLOS ONE

Journal Requirements:

Deanship of Research at Jordan University of Science and Technology - Research Grant No: 20230271. 

The authors gratefully acknowledge the Deanship of Research at Jordan University of Science and Technology for their financial support of this work (Research Grant No: 20230271). Acknowledgment is also given to the Canadian University Dubai (CUD) for approving and supervising the study design and protocol. Sincere gratitude is extended to all participants who contributed to this study

Deanship of Research at Jordan University of Science and Technology - Research Grant No: 20230271. 

Reviewers' comments:

Reviewer's Responses to Questions

**Comments to the Author**

1. Is the manuscript technically sound, and do the data support the conclusions?

Reviewer #1: Partly

Reviewer #2: Partly

2. Has the statistical analysis been performed appropriately and rigorously?

Reviewer #1: No

Reviewer #2: Yes

3. Have the authors made all data underlying the findings in their manuscript fully available?

Reviewer #1: Yes

Reviewer #2: No

4. Is the manuscript presented in an intelligible fashion and written in standard English?

Reviewer #1: Yes

Reviewer #2: No

Reviewer #1: 1. The conclusions are appropriately cautious in wording (e.g., “suggest” a high prevalence, and identifying factors as “potential risk factors”). One concern is the use of the term “prevalence” given the non-random sample; while the authors do report the proportion of their sample with DED, this might not represent the true population prevalence in all Jordanian contact lens wearers.

2. Consider a multivariate analysis (such as a logistic regression for DED presence or linear regression for OSDI score) to account for potential confounding between variables. Currently, each risk factor is examined in isolation; a multivariable approach could strengthen the evidence that certain factors are independent predictors of dry eye symptoms.

3. I noted a few minor grammar and wording issues that can be improved: for example, in the results section the word "However" is used in two consecutive sentences. There are also small typos (referring to Levene’s test as “Leven’s test”) and occasional awkward phrasing.

4. Because the sample was gathered via online convenience sampling, it may not be representative of all contact lens wearers in Jordan. The demographic skew (77% female, most under age 45) suggests a bias either in contact lens usage or survey participation. This limits how confidently one can generalize the “prevalence” beyond this surveyed group.

5. The study relies on self-reported symptoms only, without clinical examinations or objective tear film tests. This means the presence of dry eye disease is defined by symptoms alone; some participants might have clinical DED without symptoms or vice versa. The authors did use a symptom score cutoff (OSDI ≥13) to define “symptomatic DED,” which is standard, but the lack of clinical correlation is a constraint (acknowledged in the limitations).

6. Aside from the grammar/spelling issues mentioned earlier, the Results section refers to independent variables (sex, age group, etc.) as “dependent variables” – this should be corrected to avoid confusion. Also, when reporting statistical findings, it would help to provide the actual values (means, confidence intervals) for each group, not just p-values, to give readers a sense of effect size. Currently, some statements describe “highest mean OSDI scores” for certain subgroups without quantifying those means in the text.

Reviewer #2: Comments to the Editor

Dear Editor,

Thanks for the opportunity to review this manuscript. Please find my comments below.

Introduction

The introduction requires improvement in clarity and conciseness. Several sentences are overly long or repetitive, especially in the definitions and mechanisms of dry eye disease (DED). The flow could be enhanced by condensing overlapping information. Transitions between global data and local Jordanian statistics are abrupt, and lines 57–59 mention prevalence without providing concrete global examples or regional contrasts. Furthermore, the rationale for the study is not convincingly presented early in the introduction. Although the COVID-19 pandemic context is relevant, its importance is not emphasized clearly. Many citations are included, but their relevance to the study population is not well explained. A statement in the introduction promises increased awareness and reduction in contact lens-related complications, yet these aims are not mentioned when explaining the study rationale. Additionally, phrases such as “the most prevalent ocular condition” need either proper citation or softening.The study’s objective is not clearly and precisely stated.

Method

In the methods section, there is repetition, particularly in lines 100–104, where the questionnaire details are duplicated. The OSDI scoring intervals (lines 110–114) could be more clearly formatted and explained. The recruitment process lacks a step-by-step explanation. The report of a 100% participation rate is questionable and requires clarification. The method of ensuring that participants completed the questionnaire only once is not described. Using social media for convenience sampling likely introduced bias, favoring younger and more internet-active participants. This should be explicitly acknowledged. Although the OSDI tool is validated, it is not contact lens-specific. The reason for not using the CLDEQ-8 should be stated. Inclusion and exclusion criteria are too vague; for instance, terms like “active ocular disease” and “systemic medications” should be clarified with examples. The OSDI formula is inserted abruptly and would benefit from better formatting and clearer context. Confounding factors are not described, nor is there an explanation of how they were identified or controlled. The manuscript inconsistently uses correlation and association terminology—group comparisons (e.g., ANOVA) and correlational tests (e.g., Pearson) should be clearly distinguished.

Results

In the results section, percentages should be removed from Table 1 to avoid redundancy. Frequency columns should use consistent labeling such as “n (%)” and include column totals where appropriate. Statistical results like Shapiro-Wilk and Levene’s tests are overemphasized and could be summarized briefly or placed in supplementary materials. Statistical analysis descriptions should be placed in the methods section to avoid confusion. The reporting of significant results is repetitive and could be streamlined. The figures and tables are referenced, but without interpretation—readers need guidance on what these visual elements reveal. Phrasing like “recorded the highest mean score” should be replaced with “had the highest mean score” for better readability. The term “OSDI intervals” is unclear and should be replaced with “OSDI severity categories.” Avoid conflating “correlation” with “association,” particularly in categorical data. Where significant ANOVA results exist (e.g., lens age, lubricant use, mask use), post-hoc results such as Tukey HSD should be reported. Effect sizes or confidence intervals would enhance interpretation. It’s also unclear how “contact lens age” was defined—this should be clarified. Mask use data should specify whether it was self-reported and over what period. Any missing or excluded data should be reported along with how they were handled. Percentages should be written consistently, and statistical reporting should use standard notation for test statistics and p-values. Figures 1–3 use only descriptive terms without statistical interpretation. Additionally, the statement about increased screen time contributing to dryness is speculative, as screen use was not directly measured.

Discussion

The discussion repeats content already presented in the introduction and results, especially about mask use and OSDI scoring. These elements should be summarized rather than restated. Associations are sometimes incorrectly framed as causal relationships—for example, suggesting that lubricant use prevents DED, which overstates the data. The discussion includes comparisons with other studies, but when findings differ, the explanations are shallow or missing. Non-significant findings are mentioned but not explored in depth, missing the opportunity for further insight. Although the use of subheadings in the discussion is helpful, transitions between sections are abrupt and need smoothing. Redundant phrasing such as “symptomatic DED among contact lens wearers” should be reduced or varied. Environmental triggers like wind and air conditioning are reported as common but without clarification on whether these observations were statistically tested. Passive voice is used frequently and should be minimized to improve clarity.

The limitations section mentions the use of online surveys but understates the associated biases. Skewed age and gender representation should be acknowledged in more detail. The limitations of using only the OSDI tool should be addressed directly—mentioning that additional clinical measures or more specific tools like the CLDEQ-8 could have strengthened the results. Omitted risk factors, such as screen time, occupation, and smoking, are also important and should be acknowledged more explicitly.

The conclusion repeats content already discussed, with minimal added value. Instead, it should focus on the key findings, their implications for clinical care and policy, and recommendations for future research. Suggestions for future work should be concrete—for example, using clinical tests, evaluating other contact lens types, or conducting longitudinal studies.

Minor corrections

Grammatically, the manuscript requires careful revision. For instance, the phrase “an earlier study” should be clarified to specify which study is being referenced. Line 157 contains a redundant statement that should be reworded. Line 188 has a spacing error and is overly wordy. Line 167 misuses the term “confirmed the hypothesis,” which should be revised to “supported the hypothesis.”

Overall, while the topic is important and the data are relevant, the manuscript needs substantial revision before it can be considered for publication. A major revision is recommended.

**Do you want your identity to be public for this peer review?** For information about this choice, including consent withdrawal, please see our Privacy Policy

Reviewer #1: No

Reviewer #2: **Yes: ** Dr Ngozika Esther Ezinne

---

## [Author Response · Author response to Decision Letter 1]

30 Aug 2025

Reviewer #1:

1. The conclusions are appropriately cautious in wording (e.g., “suggest” a high prevalence, and identifying factors as “potential risk factors”). One concern is the use of the term “prevalence” given the non-random sample; while the authors do report the proportion of their sample with DED, this might not represent the true population prevalence in all Jordanian contact lens wearers. Response: We appreciate the reviewer’s observation. We agree that, since our sample was not randomly selected, the proportion of participants with DED in our study does not represent the true population prevalence among all Jordanian contact lens wearers. Our use of the term “prevalence” was intended descriptively; to refer to the proportion observed in our sample. To avoid any misinterpretation, we have revised the text to clarify that our results reflect the observed proportion in our study population, rather than the population prevalence, and have adjusted the wording throughout the manuscript accordingly.

2. Consider a multivariate analysis (such as a logistic regression for DED presence or linear regression for OSDI score) to account for potential confounding between variables. Currently, each risk factor is examined in isolation; a multivariable approach could strengthen the evidence that certain factors are independent predictors of dry eye symptoms. Response: a multiple regression analysis was conducted and presented in the revised manuscript.

3. I noted a few minor grammar and wording issues that can be improved: for example, in the results section the word "However" is used in two consecutive sentences. There are also small typos (referring to Levene’s test as “Leven’s test”) and occasional awkward phrasing. Response: Revised.

4. Because the sample was gathered via online convenience sampling, it may not be representative of all contact lens wearers in Jordan. The demographic skew (77% female, most under age 45) suggests a bias either in contact lens usage or survey participation. This limits how confidently one can generalize the “prevalence” beyond this surveyed group. Response: To overcome the generalizability of our findings based on bias either in contact lens usage or survey participation, the term “Prevalence” is replaced by “The proportion of DED in the study sample” We have revised the manuscript to clarify this limitation in both the Discussion and Conclusion sections and to ensure that our wording reflects that our results represent the proportion within our surveyed group rather than the true population prevalence.

5. The study relies on self-reported symptoms only, without clinical examinations or objective tear film tests. This means the presence of dry eye disease is defined by symptoms alone; some participants might have clinical DED without symptoms or vice versa. The authors did use a symptom score cutoff (OSDI ≥13) to define “symptomatic DED,” which is standard, but the lack of clinical correlation is a constraint (acknowledged in the limitations). Response: We totally agree with reviewer 1 as all the findings in this study represents the proportion of DED in the study sample of DED based on the symptoms. That’s why the term “symptomatic DED” is utilized instead of DED in all over the manuscript.

6. Aside from the grammar/spelling issues mentioned earlier, the Results section refers to independent variables (sex, age group, etc.) as “dependent variables” – this should be corrected to avoid confusion. Also, when reporting statistical findings, it would help to provide the actual values (means, confidence intervals) for each group, not just p-values, to give readers a sense of effect size. Currently, some statements describe “highest mean OSDI scores” for certain subgroups without quantifying those means in the text. Response: Revised.

Reviewer #2:

Dear Editor,

Thanks for the opportunity to review this manuscript. Please find my comments below.

Introduction

The introduction requires improvement in clarity and conciseness. Several sentences are overly long or repetitive, especially in the definitions and mechanisms of dry eye disease (DED). The flow could be enhanced by condensing overlapping information. Transitions between global data and local Jordanian statistics are abrupt, and lines 57–59 mention prevalence without providing concrete global examples or regional contrasts. Furthermore, the rationale for the study is not convincingly presented early in the introduction. Although the COVID-19 pandemic context is relevant, its importance is not emphasized clearly. Many citations are included, but their relevance to the study population is not well explained. A statement in the introduction promises increased awareness and reduction in contact lens-related complications, yet these aims are not mentioned when explaining the study rationale. Additionally, phrases such as “the most prevalent ocular condition” need either proper citation or softening. The study’s objective is not clearly and precisely stated. Response: We thank the reviewer for these constructive and detailed suggestions to strengthen the Introduction. We have revised this section to improve clarity, conciseness, and logical flow.

Method

• In the methods section, there is repetition, particularly in lines 100–104, where the questionnaire details are duplicated Response: Deleted.

• The OSDI scoring intervals (lines 110–114) could be more clearly formatted and explained Response: Revised.

• The recruitment process lacks a step-by-step explanation. Response: Revised

• The report of a 100% participation rate is questionable and requires clarification. The method of ensuring that participants completed the questionnaire only once is not described. Response: The figure reflects the proportion of respondents who completed the survey after initiating it, not the proportion of all individuals invited to participate. Since the survey was distributed via online platforms with open access, a traditional response rate based on invitations sent was not applicable. To minimize duplicate responses, we used mandatory unique identifiers where possible. We have now added a detailed explanation of these procedures in the Methods section to clarify how we addressed this issue.

• Using social media for convenience sampling likely introduced bias, favouring younger and more internet-active participants. This should be explicitly acknowledged. Response: Acknowledged in study limitations.

• Although the OSDI tool is validated, it is not contact lens-specific. The reason for not using the CLDEQ-8 should be stated. Response: We chose the OSDI because it is the only validated dry eye questionnaire currently available in Arabic, in addition to its broad applicability, established reliability, and ability to capture ocular surface symptoms relevant to both contact lens wearers and non-wearers. This also allowed for comparison with previous studies in similar populations. However, we agree that the CLDEQ-8 is a valuable contact lens–specific tool, and this important point was acknowledged in the study’s limitations.

• Inclusion and exclusion criteria are too vague; for instance, terms like “active ocular disease” and “systemic medications” should be clarified with examples. We acknowledge that some terms, such as “active ocular disease” and “systemic medications,” might have been interpreted variably by participants due to their general wording. To address this limitation, we have already clarified these terms with examples in the questionnaire, specified that such disease or medication that affect the ocular surface integrity and acknowledged the potential for variability in participant interpretation as a limitation of the study.

• The OSDI formula is inserted abruptly and would benefit from better formatting and clearer context. Response: we have revised the formula to improve readability.

• Confounding factors are not described, nor is there an explanation of how they were identified or controlled. Response: confounding factors are now described in the revised manuscript.

• The manuscript inconsistently uses correlation and association terminology—group comparisons (e.g., ANOVA) and correlational tests (e.g., Pearson) should be clearly distinguished. Response: the terms “correlation” and association” are used now correctly describing the used test.

Results

• In the results section, percentages should be removed from Table 1 to avoid redundancy. Frequency columns should use consistent labelling such as “n (%)” and include column totals where appropriate. Response: Revised

• Statistical results like Shapiro-Wilk and Levene’s tests are overemphasized and could be summarized briefly or placed in supplementary materials. Response: Moved to supplementary materials.

• Statistical analysis descriptions should be placed in the methods section to avoid confusion. The reporting of significant results is repetitive and could be streamlined. Response: The Methods section is now updated to include a clear description of the multiple linear regression analysis conducted to identify independent predictors of symptomatic dry eye severity using the continuous OSDI score. This addition clarifies how potential confounding factors were accounted for in our analysis. The revised Data Analysis subsection now explicitly details the use of multiple regression alongside the other statistical tests, enhancing the methodological transparency of our study.

• The figures and tables are referenced, but without interpretation—readers need guidance on what these visual elements reveal. Response: Revised

• Phrasing like “recorded the highest mean score” should be replaced with “had the highest mean score” for better readability. Response: Revised

• The term “OSDI intervals” is unclear and should be replaced with “OSDI severity categories.” Response: Revised

• Avoid conflating “correlation” with “association,” particularly in categorical data. Where significant ANOVA results exist (e.g., lens age, lubricant use, mask use), post-hoc results such as Tukey HSD should be reported. Effect sizes or confidence intervals would enhance interpretation. Response: Revised.

• It’s also unclear how “contact lens age” was defined—this should be clarified.

Response: In our study, “contact lens age” referred to the total duration (in months) that a participant had been wearing contact lenses since first use. We have clarified this definition in the Methods section to avoid any ambiguity.

• Mask use data should specify whether it was self-reported and over what period.

Response: Mask use data in our study were self-reported by participants through the online questionnaire. Participants were asked to report their typical mask-wearing habits during the three months preceding the survey. This clarification has been added to the Methods section.

• Any missing or excluded data should be reported along with how they were handled. Response: In our survey platform, incomplete questionnaires could not be submitted electronically; participants were required to answer all mandatory items before submission. As a result, there were no missing or partially completed data in the final dataset and all the submitted questionnaires were complete and included in the analysis.

• Percentages should be written consistently, and statistical reporting should use standard notation for test statistics and p-values. Response: Revised

• Figures 1–3 use only descriptive terms without statistical interpretation. Response: These figures were intentionally presented to provide a descriptive overview of the distribution of self-reported ocular symptoms within the study population. As these graphs depict proportions rather than comparisons between groups, no inferential statistical analyses were appropriate or required. Our interpretation was therefore limited to reporting the observed percentages, in line with the descriptive objective of these figures.

Additionally, the statement about increased screen time contributing to dryness is speculative, as screen use was not directly measured.

Response: We have revised the results/discussion to clarify that this point is based on evidence from previous literature and should be interpreted as a potential contributing factor rather than a finding from our data. Also screen time was not measured in this study this was stated clearly in the manuscript.

Discussion

The discussion repeats content already presented in the introduction and results, especially about mask use and OSDI scoring. These elements should be summarized rather than restated. Associations are sometimes incorrectly framed as causal relationships—for example, suggesting that lubricant use prevents DED, which overstates the data. The discussion includes comparisons with other studies, but when findings differ, the explanations are shallow or missing. Non-significant findings are mentioned but not explored in depth, missing the opportunity for further insight. Although the use of subheadings in the discussion is helpful, transitions between sections are abrupt and need smoothing. Redundant phrasing such as “symptomatic DED among contact lens wearers” should be reduced or varied. Environmental triggers like wind and air conditioning are reported as common but without clarification on whether these observations were statistically tested. Passive voice is used frequently and should be minimized to improve clarity. Response: We have revised and updated the discussion section to make it more concise and informative.

The limitations section mentions the use of online surveys but understates the associated biases. Skewed age and gender representation should be acknowledged in more detail. The limitations of using only the OSDI tool should be addressed directly mentioning that additional clinical measures or more specific tools like the CLDEQ-8 could have strengthened the results. Omitted risk factors, such as screen time, occupation, and smoking, are also important and should be acknowledged more explicitly. Answer: The limitation section is now updated.

The conclusion repeats content already discussed, with minimal added value. Instead, it should focus on the key findings, their implications for clinical care and policy, and recommendations for future research. Suggestions for future work should be concrete—for example, using clinical tests, evaluating other contact lens types, or conducting longitudinal studies. Response: Revised.

Minor corrections

Grammatically, the manuscript requires careful revision. For instance, the phrase “an earlier study” should be clarified to specify which study is being referenced. Answer: Revised.

Line 157 contains a redundant statement that should be reworded. Answer: Revised.

Line 188 has a spacing error and is overly wordy. Response: Revised.

Line 167 misuses the term “confirmed the hypothesis,” which should be revised to “supported the hypothesis.” Response: Revised

Overall, while the topic is important and the data are relevant, the manuscript needs substantial revision before it can be considered for publication. A major revision is recommended. Thank you for your valuable feedback

---

## [Decision Letter · Decision Letter 1]

8 Oct 2025

Symptomatic dry eye disease (DED) in cohort of contact lens wearers in Jordan

PONE-D-25-34122R1

Dear Dr. Ghach,

We’re pleased to inform you that your manuscript has been judged scientifically suitable for publication and will be formally accepted for publication once it meets all outstanding technical requirements.

Kind regards,

Clara Martínez Pérez

Academic Editor

PLOS ONE

Additional Editor Comments (optional):

Reviewers' comments:

Reviewer's Responses to Questions

**Comments to the Author**

Reviewer #1: All comments have been addressed

Reviewer #2: All comments have been addressed

2. Is the manuscript technically sound, and do the data support the conclusions?

Reviewer #1: Yes

Reviewer #2: Yes

3. Has the statistical analysis been performed appropriately and rigorously?

Reviewer #1: Yes

Reviewer #2: No

4. Have the authors made all data underlying the findings in their manuscript fully available?

Reviewer #1: Yes

Reviewer #2: No

5. Is the manuscript presented in an intelligible fashion and written in standard English?

Reviewer #1: Yes

Reviewer #2: Yes

Reviewer #1: All comments have been addressed which include concerns on typos, use of multivariate analysis, correction to the independent variables, and the misuse of the term prevalence.

Reviewer #2: Thank you for the thoughtful revision—clarity and organization have improved. I have several additional suggestions to strengthen the manuscript further.

Recruitment is via social-media convenience sampling, yet parts of the text describe “random distribution.

Symptomatic DED defined by OSDI ≥13 is acceptable, but the manuscript should consistently cite the Arabic OSDI validation and justify the cut-points. Prevalence proportion(s) should include 95% CIs overall and by key strata.

The primary model is a multiple linear regression of OSDI scores with R²=0.828 while all predictors have VIF>5 and tolerance <0.2. This implies severe multicollinearity/overfitting, undermining inference. Several predictors also overlap conceptually (mask use, mask frequency, mask+CL interaction; “CL_symptoms” predicting a symptom scale may be tautological). As is, the adjusted associations are not reliable. Given OSDI’s bounded distribution and the frequent use of categories, a sensitivity analysis using ordinal logistic regression for OSDI categories would strengthen robustness.

Numerous univariable tests (ANOVA, χ², correlations) are presented without multiplicity control. Either limit inferential emphasis to the multivariable model (reporting effect sizes with CIs) or apply FDR control. With n=301, reliance on Shapiro–Wilk p=0.081 to claim normality of OSDI is weak; OSDI is bounded (0–100) and often right-skewed. Robust SEs and visual diagnostics (Q–Q, scale-location) are more appropriate. The reported residual diagnostics are minimal.

Mask variables (binary, frequency, concurrent CL use) are likely correlated with pandemic-period behaviors and other confounders (screen time, indoor AC exposure), which were not measured. Causal language should be tempered; associations are cross-sectional and confounded.

Abstract states “24.87% were soft contact lens wearers,” yet the text says most wore soft lenses. This discrepancy suggests data/typing errors that need resolving. The abstract over-weights ANOVA results; emphasize adjusted findings (with CIs) instead.

Including a variable “CL_symptoms” as a predictor of OSDI (a symptom scale) risks circularity. This should be removed or justified; otherwise, the model inflates explanatory power.

STROBE adherence: key items are missing or under-reported (sampling frame detail, missing data handling, sensitivity analyses, participant flow, sample-size rationale/power).

The MADE discussion should acknowledge the directionality limits and potential residual confounding (e.g., screen time, indoor climate, ocular surface comorbidities).

Ensure consistent terminology (“soft contact lenses” vs “rigid gas permeable”; “mask use with CL”; “cleaning frequency”).

Recommendation

Minor Revision. The topic is suitable and regionally valuable, but publication should be contingent on substantial methodological and reporting improvements.

Specific Requests to Authors

1. Clarify sampling (convenience via social media), remove references to “random,” and temper generalizability/prevalence claims across the manuscript (title/abstract/results/discussion).

2. Correct data inconsistencies (e.g., % soft lens users) and provide a table of lens types with clear denominators.

3. Apply multiplicity control (e.g., Benjamini–Hochberg) or explicitly restrict inference to pre-specified outcomes/predictors.

4. Provide prevalence/proportion estimates with 95% CIs (overall and by key strata).

5. Temper causal language regarding mask use and lubricants; frame as associations and note unmeasured confounding (screen time, indoor humidity, lens materials).

6. Adhere to STROBE, including sample-size rationale

7. Language/formatting edit for clarity and consistency.

**Do you want your identity to be public for this peer review?** For information about this choice, including consent withdrawal, please see our Privacy Policy

Reviewer #1: No

Reviewer #2: No

---

## [Editor Report · Acceptance letter]

PONE-D-25-34122R1

PLOS ONE

Dear Dr. Ghach,

I'm pleased to inform you that your manuscript has been deemed suitable for publication in PLOS ONE. Congratulations! Your manuscript is now being handed over to our production team.

Kind regards,

on behalf of

Dr. Clara Martínez Pérez

Academic Editor

PLOS ONE